# Statistically Indistinguishable, Operationally Distinct:
# A Formal Barrier for Tabular Foundation Models

**Tassilo Klein** [1]   **Johannes Hoffart** [1]

## Abstract

Tabular foundation models cannot reason about data produced by running systems without access to the rules that govern them. We make this statement falsifiable. The *Operational Turing Test* (OTT) constructs pairs of legal and rule-violating database states whose 1- and 2-way column-value marginals match to a total variation of $< 0.02$; Le Cam's lemma then bounds any values-only classifier at $\geq 0.49$ Bayes error. Three values-only baselines (XGBoost, TabICL, TabPFN) hit the bound exactly (accuracy $0.50$, pre-registered two one-sided tests (TOST) $p < 0.002$), raw row-level access does not help, exposing relational value consistency closes most of the gap, and only a classifier fed by seven executable rule-derived audits reaches $1.00$ classification accuracy. In three matched 100-state frontier large-language-model (LLM) runs, models given the schema, trigger source, rule tables, and state files classify at most $2/50$ legal states as LEGAL; GPT-5.5 accepts $0/50$ legal states even with higher reasoning effort and a Structured Query Language (SQL) executor. The access-ladder pattern also appears on a second schema with structurally distinct rule families (banking ledger: cross-row balance, cumulative aggregate). The barrier is identifiability, not capacity: scale, data, and richer features cannot cross it without operational grounding.[1]

## 1  Introduction

Tabular foundation models have made rapid progress on benchmarks of single, self-contained tables. Production tables are different: application code writes rows, declarative constraints restrict values, procedural triggers enforce transi-

[1]SAP SE. Correspondence to: Tassilo Klein <tassilo.klein@sap.com>.

*Proceedings of the 2nd ICML Workshop on Foundation Models for Structured Data, Seoul, South Korea. 2026.*

[1]Code and data: github.com/SAP-samples/operational-turing-test.

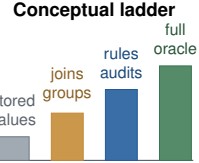 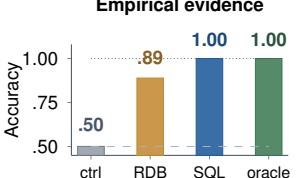

*Figure 1.* **Access ladder and empirical validation.** Left: conceptual access levels. Right: representative evidence; the dashed line is chance. Leakage controls stay at chance; relational baselines (HistGB/RDB-PFN, Wang et al., 2026) reach $0.89$ but miss derivation; SQL audits derived from schema and code match the oracle.

tions, and business rules decide which states are legal. The training data available to models is therefore a projection of a running system: stored values remain, but much of the operational logic and procedure that generated them is left behind.

This decoupling is severe in enterprise systems. Large schemas expose thousands of tables with technical identifiers whose semantics live outside the data definition language (DDL); column names such as `BUKRS`, `MATNR`, or `VBAK-ERDAT` are typical enough that automated column-name expansion is now studied as a pre-condition for working with such tables (Zhang et al., 2023). Recent cross-table (Kim et al., 2024), relational (Fey et al., 2024; Robinson et al., 2024; Wang et al., 2025), and database-native (Wehrstein et al., 2025) foundation models make the relational layer ingestible. The operational layer above it is less developed: value domains and referential integrity in schema text, plus procedural rules in triggers, validation scripts, and application code. This paper makes the resulting indistinguishability barrier falsifiable and tests whether source artifacts in a frontier LLM prompt close it.

**Reading the ladder.** Figure 1 is not a leaderboard over model classes. The gray bar is a leakage-control sanity check: values-only models and raw-row variants should remain at chance when matched states preserve observable value statistics. Relational features add joins, key coverage, and group counts; rule-derived audits instead recompute legality predicates such as stored totals from line items, discounts, and taxes.

**The Operational Turing Test.** We treat "operational context" concretely as the set of rules that decide whether a

database state is legal: value-domain constraints, value-transformation logic, relational value consistency (referential integrity), and value-state-transition constraints. The *Operational Turing Test* (OTT) generalizes this idea to tests where statistical observations are insufficient without operational rules (Klein & Hoffart, 2026); here we instantiate it as a binary classification task on pairs of (legal, illegal) states where the rule-violating corruption preserves 1- and 2-way column-value marginals to within total variation distance (TV) $\tau = 0.02$. Le Cam's lemma gives a Bayes error bound of $\frac{1}{2} - \frac{\tau}{2} \geq 0.49$ for any classifier whose input is restricted to such statistics (Proposition 1). Values-only baselines from two broad architectural families, gradient-boosted trees and transformers, all reach accuracy statistically equivalent to chance under a pre-registered two one-sided tests (TOST) procedure for equivalence[2] (Lakens, 2017, $\alpha = 0.05$); seven executable rule-derived audits close the gap to 1.00 classification accuracy.

**Findings.**

1. Operational grounding is **information-theoretically necessary** under marginal matching (Proposition 1).
2. The barrier is **architecture-, scale-, and representation-independent**: three baselines and raw rows all hit the Le Cam bound (Section 4).
3. Exposing relational value consistency closes most of the gap but **cannot recover value-transformation logic** (Table 1).
4. Frontier reasoning models with full source artifacts in-prompt **fail the test** (Section 5).

## 2 The Operational Turing Test

**Setting.** Real-world tables do not stand alone; they are rows written by a running system, and the rules governing that system come in two kinds. *Declarative rules* (types, value-domain constraints, and relational value consistency such as referential integrity) are written into the schema text. *Procedural rules* (value-state-transition constraints, value-transformation logic, and per-aggregate cardinality bounds) live in trigger and application code. We write the union of both as a rule set $\Pi$. A dataset stripped of $\Pi$ retains the values but loses much of the operational logic and procedure that produced them.

Write $\phi_k(S)$ for the vector of all $k$-way column-value marginals of database state $S$, and call a classifier *values-only* if its input is $\phi_k(S)$ for some finite $k$ (so it has no access to queries that read $\Pi$). Let $P_{\text{legal}}$ and $P_{\text{illegal}}$ be the distributions over states from a legal data-generating process and from a corruption that violates a single rule of $\Pi$ while preserving values elsewhere.

**Proposition 1** (Necessity of operational grounding). *If* $\|\phi_k(P_{legal}) - \phi_k(P_{illegal})\|_{\text{TV}} \leq \tau$, *then any values-only classifier has Bayes error* $\geq \frac{1}{2} - \frac{\tau}{2}$.

This is Le Cam's lemma (Le Cam, 1973; Tsybakov, 2009): two distributions within $\tau$ in total variation cannot be distinguished better than $\frac{1}{2} - \frac{\tau}{2}$ Bayes error. Our construction enforces $\tau < 0.02$ (Section 3), giving $\geq 0.49$. The bound is an *existence/impossibility* result. The failure is identifiability-based, not capacity-based, so larger models, more data, and richer features cannot escape it; we verify the last empirically with row-level access (Section 4). We do not claim every tabular foundation model fails on every task, only that where this condition holds, rule-guided queries are necessary.

## 3 Construction

**Schema.** We instantiate the test on a three-table order-to-cash schema (`customers`, `orders`, `order_items`), the structural core of a standard enterprise sales process. The schema is deliberately minimal for transparency; the full schema definition together with the trigger and application-code rules is in Appendices A.1 to A.3, and the construction generalises to any schema whose rules can be stated as executable code. Real-world linked business-table corpora at scale are already available (Klein et al., 2024).

**Rules and corruptions.** Four families of operational rules govern legal states; only some live in the schema text. For each we give location, instance, and a single-rule corruption that preserves every 1- and 2-way column-value marginal to $\tau < 0.02$:

(i) *Relational value consistency* (declarative, schema). `orders.customer_id` must occur in `customers.id`. `fk_break` replaces $\sim$5% of values with unseen IDs. Surrogate key symbols are excluded from $\phi_k$ to make the values-only view invariant to arbitrary ID renamings; checking whether a child key occurs in the parent table is instead an executable relational audit, so TV $= 0$ for the values-only view.

(ii) *Cardinality consistency* (mixed; per-row `quantity`$> 0$ in schema, per-aggregate bounds in app code). $\leq$ 20 items per order, $\leq$ 3 open orders per customer. `cardinality_break` relocates items between same-customer orders, leaving `quantity` and `unit_price` untouched; order totals are recomputed to remain consistent with the relocated items, so the value-transformation logic continues to hold.

(iii) *Value-transformation logic* (procedural, app code). `line_total = qty · unit_price · (1 − discount(tier, qty))` and `order.total = (∑ line_total) · (1 + tax(country))`. `derivation_break` adds small offsets to

`orders.total` that cancel out across orders ($\sim$0.1% of column mean, summing to zero), so the column histogram is essentially unchanged (TV $\approx 0.001$–$0.003$) but the totals no longer match the formula.

(iv) *Value-state transition* (procedural, trigger code; Appendix A.2). (`prev_status`, `status`) must lie in a four-element allowed set (e.g. `pending`→`shipped`). `transition_break` swaps `status` between one `delivered` and one initial `pending` order; the 1-way marginal of `status` is preserved exactly.

All four corruptions pass a permutation-null check ($p = 1.000$); the corruption strategies are summarized in Appendix A.4.

**Features.** The *values-only* feature vector is a 35-dimensional summary of column-level statistics: mean, standard deviation, quartiles, entropy, mode frequency, and two pre-specified cross-column correlations. By design it excludes any group-size statistic, verified invariant under permutation of `order_id` (max difference $= 0$). The *operationally-grounded* classifier augments this vector with the scalar outputs of seven executable rule-derived audits (listed in full in Appendix A.5): orphan-FK counts (two), cardinality-violation counts (two), maximum absolute residual of the line-total and order-total value transformations (two), and illegal-transition count. For value-transformation logic, the audit does not read a hidden violation label: it recomputes expected line and order totals from quantities, prices, discounts, and taxes, then compares them with the stored totals. Each audit is a single SQL query against the schema together with the trigger or application code; a model without access to $\Pi$ cannot construct it.

**Values-only sanity baselines and scale.** We test three values-only baselines across two broad architectures: gradient-boosted trees, represented by XGBoost (Chen & Guestrin, 2016), and transformer-based tabular models, represented by TabICL (Qu et al., 2025) and TabPFN v1 (Hollmann et al., 2023). These baselines check for leakage, finite-sample artifacts, and representation effects within the values-only regime; under Proposition 1 they are not expected to solve the task. Improvements within that regime (TabPFN's Nature release (Hollmann et al., 2025a), TabPFN-2.5 (Hollmann et al., 2025b)) cannot escape it; we use v1 because the failure mode is information-theoretic, not capacity-bound. An operationally-grounded XGBoost uses the same 35-dim vector augmented with the outputs of seven executable audits; an oracle executes the rules directly and serves as an upper bound. Training: 1,000 labeled (legal, illegal) pairs per seed; testing: 500 held-out pairs; 5 seeds; 200 customers per state.

## 4 Results

> **Findings.**
> - **Architecture-independent failure**: XGBoost, TabICL, and TabPFN all at chance under TOST ($p < 0.002$).
> - **Representation-independent failure**: row-level access also at $0.50$ on all four violations.
> - **Relational value consistency is insufficient**: tree and PFN models with relational features miss value-transformation logic at recall $\leq 0.02$.
> - **Executable rule-derived audits close the gap**: operationally-grounded and oracle reach $1.00$.
> - **LLMs given the source artifacts**: matched 100-state runs predict at most $2/50$ legal states as LEGAL.

**Chance across architectures.** All three values-only models are statistically equivalent to chance within $\pm 2$ pp under a pre-registered TOST equivalence test (Lakens, 2017, $\alpha = 0.05$): XGBoost **0.5014** ($p = 0.0018$), TabICL **0.5006**, TabPFN **0.5012** (both $p < 10^{-4}$); see Figure 1. The operationally-grounded XGBoost reaches **1.00** and the oracle 1.00. Per-violation recalls for the values-only models are consistent with random guessing, and the errors of the two architecturally most distant baselines (XGBoost, TabICL) are essentially uncorrelated (Pearson $\varphi = -0.076$), ruling out a shared exploited signal. TabICL further outputs $P(\text{legal}) = 0.5001 \pm 0.0029$ on *every* test instance: maximal epistemic uncertainty (Hüllermeier & Waegeman, 2021; Melo et al., 2026) in response to an information-theoretic void, rather than guessing.

**Scale and representation do not help.** Values-only accuracy is **flat at 0.50** from 50 to 5,000 training pairs (the operationally-grounded model saturates near 1.00 from just 50), consistent with Proposition 1 being information-theoretic, not statistical. Raw row-level access fares no better: TabICL on `orders` rows (per-row predictions, majority vote) reaches **0.50** on all four violations, including the within-row `transition_break`. Without rule context to direct attention, raw rows provide the data but not the query.

**Relational value consistency helps but does not close the gap.** A tree with relational features (cross-table joins, FK coverage, per-group degrees, and the within-row (`prev_status`, `status`) value pair) reaches **0.89** (HistGB, 3 seeds), recovering FK, cardinality, and value-state transitions at recall 1.00 but missing value-transformation logic entirely (Table 1). Value-state transition is informative here: although the legal transition set is encoded only in trigger code, an illegal pair never appears in the legal training states, so a single occurrence in a test state is enough for a tree to flag once the (`prev_status`, `status`) pair is exposed as a feature col-

umn. The 35-dim values-only vector excludes that pair, which is why the gap between values-only and relational is mostly the addition of within-table two-column value-pair features, not anything cross-table. A relational PFN (Wang et al., 2026) reproduces the same pattern. What no relational tier recovers is *value-transformation logic*: tax and discount formulas cannot be reconstructed from connectivity. Probabilistic relational models (Hilprecht et al., 2020) capture statistical structure but not the executable formulas that produced it; that residual gap is closed only by reading the rule code.

| Access tier | FK | Card. | Transform. | State |
|---|---|---|---|---|
| Values-only (XGB) | 0.54 | 0.50 | 0.49 | 0.54 |
| Row-level (TabICL) | 0.50 | 0.50 | 0.50 | 0.50 |
| Relational (HistGB) | 1.00 | 1.00 | **0.00** | 1.00 |
| Relational (RDB-PFN) | 1.00 | 1.00 | **0.02** | 1.00 |
| Audit-features (XGB) | 1.00 | 1.00 | 1.00 | 1.00 |

*Table 1.* **Per-violation recall across the access ladder.** Relational features perfectly recover FK, cardinality, and value-state-transition violations, but *neither* relational baseline recovers value-transformation logic. Executable rule-derived audits close the remaining gap.

**Alternative explanations ruled out.** *Weak classifier?* Three architectures across distinct inductive biases fail identically. *Aggregate features unfair?* Row-level access also at 0.50 on all four violations. *Just need relational structure?* Tree and PFN models with full relational features still miss value-transformation logic (recall $\leq 0.02$). *Training artifact?* Flat scaling over two orders of magnitude. *Feature leakage?* Values-only vector invariant to order_id permutation. *Near-zero mutual information exploited stochastically?* The observed feature-label mutual information sits at the bottom of a label-shuffled null distribution ($z = -1.86$, pooled $p = 0.999$), so even the trace amount that exists is no larger than chance.

## 5  LLMs With the Rule Source in the Prompt

A natural objection is that frontier reasoning models, given the schema and rule code, can read and execute the rules. We tested this directly, and it does not hold here. Kimi-K2.6 (Moonshot AI Team, 2026) and GPT-5.5 (OpenAI, 2026) each received the schema, trigger source, rule tables, and state files in one prompt. In Table 2, all rows use the same 100-state evaluation set (50 legal, 50 illegal), making the key diagnostic comparable: how often a truly legal state is classified as LEGAL.

False positives concentrate on a small set of violation types. GPT-5.5 at high reasoning effort marks every legal state as transition_break; Kimi-K2.6 accepts only 2/50 legal states and otherwise emits a violation label or EMPTY response. This is not a claim that LLMs are useless in an operational harness. Rather, LLMs *on their own*, even with source artifacts in context, do not reliably compile those

| Run | N | Acc. | LEGAL on legal |
|---|---|---|---|
| GPT-5.5 default prompt | 100 | 0.50 | 0/50 |
| GPT-5.5 high effort | 100 | 0.50 | 0/50 |
| GPT-5.5 + SQL executor | 100 | 0.50 | 0/50 |
| Kimi-K2.6 | 100 | 0.46 | 2/50 |

*Table 2.* **Frontier reasoning models with full source artifacts.** Overall accuracy is secondary because the dominant failure mode is asymmetric: models often detect illegal states while rejecting legal ones. The diagnostic count is therefore LEGAL predictions on truly legal states. Kimi-K2.6 produced EMPTY responses on 14/100 states, including 8/50 legal states.

artifacts into the finite checks that decide legality.

The follow-ups rule out simple resource explanations. Kimi-K2.6 has an additional output-cap failure mode (14/100 EMPTY responses, including 8 legal states), but among non-empty legal responses it still accepts only 2/42 legal states; a higher-budget diagnostic accepts only 1/25. GPT-5.5 has no EMPTY responses; at high reasoning effort it labels every illegal state correctly, with 41/50 correct violation types, yet still classifies 0/50 legal states as LEGAL. In the SQL-executor variant, the model can run queries it writes, but the audit is still specified ad hoc by the model rather than compiled from the rules. It often queries relevant tables and columns, but the generated checks are partial or misspecified: the model misstates the legality predicate (for example the initial-state convention for prev_status) and misses value-transformation checks (0/17). The failure is therefore specification, not arithmetic or SQL execution. A harness that compiles schema, trigger code, and rule tables into deterministic audits is exactly the operational grounding hypothesis; those audits over the same artifacts reach 1.00.

## 6  Discussion & Conclusion

Operational grounding is required in the setting studied here. For data produced by a running system, there exist operationally realistic instances on which no model without executable rule logic can do better than random chance, whether it sees aggregate features or raw rows. The failure is identifiability-based: scaling cannot resolve it, and the binding constraint is the absence of executable logic, not the choice of representation. Single-table benchmarks like TabArena (Erickson et al., 2025) miss this failure mode; relational benchmarks (RelBench, Robinson et al., 2024) expose part of it; only operational benchmarks expose all of it. Progress therefore means treating declarative and procedural rules as first-class inputs: not as retrieved text, but as logic the model can evaluate (Meta FAIR CodeGen Team, 2025). For deployed agents, the relevant capability is not only table prediction but operational verification: knowing which rule must be evaluated before a state can be trusted.

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

# A    Schema and Example Data

This appendix exhibits the schema definition, complementary rule code, executable rule set $\Pi$, and corruption strategies. The construction is fully reproducible; code is released at github.com/SAP-samples/operational-turing-test. Figure 2 visualises the relational structure and the operational rules layered on top of it.

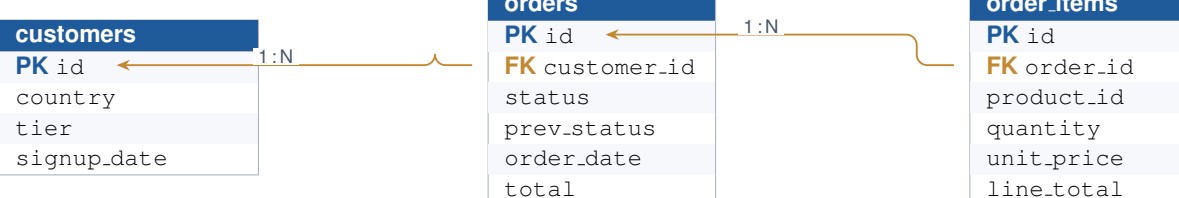

*Figure 2.* **Database schema.** Three tables linked by foreign keys (**FK** → **PK**). Arrows point from each FK column directly to the PK column it references: `orders.customer_id` → `customers.id` and `order_items.order_id` → `orders.id`, each a one-to-many relation. The four operational-rule families layered on top of this structure (referential integrity, cardinality, derivation, transition) are detailed in the appendix; only the first is fully declared in the schema text, the others live partially or entirely in trigger and application code.

## A.1    Schema

The three-table order-to-cash schema. The schema declares only the declarative part of $\Pi$: foreign keys (rule family i), value-domain CHECKs, and the per-row `quantity > 0` bound (part of family ii). The remaining rules live in trigger and application code (next subsection): the per-customer open-order limit (the rest of family ii), the line- and order-total value-transformation formulas (family iii), and the legal value-state-transition pairs (family iv). The `prev_status` and `total` columns appear without any CHECK constraining their joint or derived values; both invariants are enforced procedurally.

```
CREATE TABLE customers (
    id           INTEGER PRIMARY KEY,
    country      VARCHAR(2),
    tier         VARCHAR(10) CHECK (tier IN ('bronze','silver','gold')),
    signup_date  DATE
);

CREATE TABLE orders (
    id           INTEGER PRIMARY KEY,
    customer_id  INTEGER REFERENCES customers(id),
    status       VARCHAR(20)
                 CHECK (status IN ('pending','shipped','delivered','cancelled')),
    prev_status  VARCHAR(20),
    order_date   DATE,
    total        NUMERIC(10,2)
);

CREATE TABLE order_items (
    id           INTEGER PRIMARY KEY,
    order_id     INTEGER REFERENCES orders(id),
    product_id   INTEGER,
    quantity     INTEGER CHECK (quantity > 0),
    unit_price   NUMERIC(10,2),
    line_total   NUMERIC(10,2)
);
```

The schema declares column types, primary/foreign keys, and value-domain CHECKs, but is silent on which (`prev_status`, `status`) pairs are legal, on the per-customer open-order limit, and on how `line_total` and `total` are derived. A model given the schema text alone cannot construct an audit query for any of those rules.

## A.2    Procedural Rule (Trigger Code)

The legal value-state-transition set is enforced by a trigger function rather than a CHECK constraint, consistent with how multi-row temporal rules are typically expressed in enterprise systems. The transition rule is therefore visible only to a model with access to the trigger source:

```
CREATE OR REPLACE FUNCTION check_status_transition()
RETURNS TRIGGER AS $$
DECLARE
    allowed CONSTANT TEXT[][] := ARRAY[
        ['pending', 'shipped'],
        ['shipped', 'delivered'],
```

```
            ['pending',  'cancelled'],
            ['pending',  'pending']     -- initial state
    ];
BEGIN
    IF NOT (ARRAY[NEW.prev_status, NEW.status] = ANY (allowed))
    THEN
        RAISE EXCEPTION 'illegal transition % -> %',
            NEW.prev_status, NEW.status;
    END IF;
    RETURN NEW;
END;
$$ LANGUAGE plpgsql;

CREATE TRIGGER orders_status_transition
BEFORE INSERT OR UPDATE ON orders
FOR EACH ROW EXECUTE FUNCTION check_status_transition();
```

The same pattern (some rules in the schema, others in trigger or application code) holds for the per-customer cardinality bound and the `discount`/`tax` value-transformation formulas, which we omit here for brevity but treat identically: they enter the operational rule set $\Pi$ alongside the schema-level constraints.

### A.3 Operational Rules ($\Pi$)

Together, the schema constraints and the trigger/application code above encode four families of operational rules. Their intersection partitions the database state space $\Omega$ into legal ($\bigcap_i \pi_i$) and illegal regions:

1. **Relational value consistency** (in schema). Every `orders.customer_id` occurs in `customers.id`; every `order_items.order_id` occurs in `orders.id`.
2. **Cardinality consistency** (mixed). Every order has between 1 and 20 line items; every customer has at most 3 open orders, where *open* $\equiv$ `status` $\in$ {`pending`, `shipped`}. The per-customer bound is enforced in application code, not in the schema text.
3. **Value-transformation logic** (in application code). `line_total` $=$ `qty` $\cdot$ `unit_price` $\cdot$ $(1 -$ `discount(tier, qty)`), and `order.total` $= \left( \sum \text{line\_total} \right) \cdot (1 + \text{tax(country)})$. The `discount` and `tax` functions are deterministic table lookups (e.g., `discount(gold, ≥1)` $=0.10$; `tax(DE)` $=0.19$).
4. **Value-state transition** (in trigger code; see Appendix A.2). The pair (`prev_status`, `status`) lies in the allowed transition set, with (`pending`, `pending`) as the initial state.

### A.4 Corruption Strategies

Each corruption modifies a legal database state to violate exactly one rule while preserving the measured 1- and 2-way column-value marginals to within $\tau < 0.02$. `fk_break` replaces some parent keys with non-existing identifiers. `cardinality_break` redistributes line items within a customer so a group-size bound is violated while row values are preserved. `derivation_break` adds paired zero-sum perturbations to stored totals, leaving histograms essentially unchanged but breaking the discount/tax value-transformation formula. `transition_break` swaps status values so 1-way status counts are preserved while (`prev_status`, `status`) pairs leave the trigger-defined allowed set.

### A.5 The Seven Executable Audits

The operationally-grounded classifier in Section 4 uses the scalar outputs of seven executable rule-derived audits computed directly from the schema and the trigger / application code above. Each is a one-line audit query:

1. `n_orphan_fk_orders`: count of `orders` rows with `customer_id` $\notin$ `customers.id`.
2. `n_orphan_fk_items`: count of `order_items` rows with `order_id` $\notin$ `orders.id`.
3. `n_orders_over_item_limit`: count of orders with more than 20 line items.
4. `n_cust_over_open_limit`: count of customers with more than 3 open orders.
5. `max_abs_lt_residual`: $\max_i |\text{line\_total}_i - \text{qty}_i \cdot \text{unit\_price}_i \cdot (1 - \text{discount(tier, qty)})|$.
6. `max_abs_order_residual`: $\max_o |\text{order.total}_o - \left( \sum \text{line\_total} \right) \cdot (1 + \text{tax(country)})|$.
7. `n_illegal_transitions`: count of (`prev_status`, `status`) pairs not in the allowed set.

Each value is a single SQL query against the database. A model with access to both schema and rule code can derive them; a model without that access cannot. This is what we mean by "operational grounding."

