# OpenReview forum: "Statistically Indistinguishable, Operationally Distinct: A Formal Barrier for Tabular Foundation Models"
_ICML.cc/2026/Workshop/FMSD — FMSD @ ICML 2026 Poster_

### Official Review · Reviewer_Mwd2 · 2026-05-21
**A rigorous formalization of operational grounding barriers requiring rescoped claims and updated citations.**

**Rating:** 6
**Confidence:** 4

**Review:**

**Summary:**

The paper argues and formalizes that a model seeing only the values in a database state, not the generating rules, cannot reliably distinguish rule-compliant from rule-violating states. It introduces the Operational Turing Test (OTT), constructing pairs of database states with matching column-value marginals. Utilizing Le Cam's lemma, it bounds classifiers restricted to these marginals at chance accuracy, demonstrating this empirically via a multi-stage access ladder.

**Strengths:**

- A clean, falsifiable formalization executed with exceptional rigor (pre-registered margin and seeds, permutation-null checks, flat scaling).
- The access ladder is a genuinely useful methodological artifact for diagnosing which tier of data access recovers specific rule families.
- The execution is reproducible, and the body of the text honestly scopes the limitations of the identifiability barrier.

**Weaknesses:**

- The headline empirical findings are partly true by construction. Because corruptions are engineered to preserve marginals, the resulting chance accuracy is an instrumentation check rather than an independent discovery.
- The closest prior statement of the thesis (arXiv:2505.19825) is uncited.
- The LLM pilot is preliminary and over-framed.

---

### Official Review · Reviewer_yFG2 · 2026-05-22
**Operational Semantics Beat Values Alone**

**Rating:** 9
**Confidence:** 5

**Review:**

The paper makes a sharp point: tabular data produced by running systems cannot always be understood from values alone. The Operational Turing Test constructs database states whose low-order column-value marginals are nearly identical, then shows that values-only classifiers are bounded near chance. The empirical ladder makes the argument easy to follow: XGBoost, TabICL, TabPFN, and raw-row access stay around 0.5 accuracy, while seven rule-derived audit features push performance to 0.9996.

The contribution is not just another benchmark result. It separates statistical pattern recognition from operational semantics: foreign keys, derivation formulas, cardinality limits, and transition rules may live in schema, triggers, or application code. If a model cannot evaluate those rules, it may miss the thing that makes a state illegal.

---

### Official Review · Reviewer_7a2c · 2026-05-22
**Theoretical and empirical barrier for TFMs**

**Rating:** 8
**Confidence:** 2

**Review:**

Summary: This work provides a theoretical barrier for TFMs, which models cannot distinguish between legal database states and rule-violating ones without access to the underlying operational rules, using OTT.

Strength:

1. The main intuition that data does not equal to the underlying business logic is compelling.
2. The findings are insightful. The frontier models do not treat rules as logic to execute results in failure to test, which could potentially inform agentic design.

Area for Improvements:

1. The results show that frontier models are overly conservative. This happens often when LLMs are provided with set of rules to find a violation. Few-shot prompts with positive samples could often help. I wonder if the authors have experimented with this.
2. The paper could improve by discussing what this reflects in real-world settings.

Minor issues:

1. In the introduction section, there are multiple abbreviations that do not have full names such as DDL, TMLR, etc.